# The Relationship between Distress Tolerance and Spiritual Well-Being towards ARV Therapy Adherence in People Living with HIV/AIDS

**DOI:** 10.3390/healthcare12080839

**Published:** 2024-04-16

**Authors:** Ramal Saputra, Agung Waluyo, Chiyar Edison

**Affiliations:** Faculty of Nursing, Universitas Indonesia, Depok 16424, Indonesia; ramal.suyono@alumni.ui.ac.id (R.S.); c.edison@ui.ac.id (C.E.)

**Keywords:** antiretroviral, distress tolerance, spiritual well-being

## Abstract

A crucial factor in the success of treatment for patients with Human Immunodeficiency Virus (HIV) is adherence to antiretroviral (ARV) therapy among People Living with HIV/AIDS (PLWHA). Adherence issues remain a persisting problem with multifaceted causes. There are many studies on variables related to ARV therapy adherence, but no study has been found on spiritual well-being and distress tolerance in ARV therapy adherence. This study aims to determine the relationship between distress tolerance and spiritual well-being on adherence to ARV therapy in PLWHA. This research used a quantitative approach with a cross-sectional design. The sample collection process followed a consecutive sampling technique, with data gathered from 129 participants at the South Lampung Regional General Hospital located in Indonesia. Data collection was conducted using three questionnaires administered by the interviewer, which assessed distress tolerance using the Miller–Smith Rating Scale For Stress Tolerance (MSRS-ST), evaluated spiritual Well-Being using the Spiritual Well-Being Scale (SWBS), and gauged ARV therapy adherence using the Medication Adherence Rating Scale (MARS). Data analysis using a simple logistic regression with a 95% confidence interval (CI) showed a significant relationship between distress tolerance (*p*-value 0.002) and spiritual well-being (*p*-value 0.036) towards ARV therapy adherence in PLWHA. The results of multiple logistic regression yielded distress tolerance as the most dominant and influential variable in this research. Distress tolerance and spiritual well-being impact adherence to ARV therapy in PLWHA. Suggestions for healthcare services should consider these factors to decrease the risk of non-adherence to therapy and inadvertently heighten mortality risk.

## 1. Introduction

In the context of the health crisis, HIV requires a response from the populace and calls for preventive and curative services for individuals diagnosed with the disease. These align with the Millennium Development Goals (MDGs), which comprise the reduction in HIV and other communicable and infectious diseases [1]. In Indonesia, discrimination against People Living with HIV/AIDS (PLWHA) remains at a high level [2]. The responsibilities of healthcare providers include socialization, which leads to the development of new beliefs and professional attitudes. This role focuses on areas related to health promotion and disease prevention [3,4,5,6], as well as the influence of adherence to clinical treatment [7,8,9].

According to the World Health Organization (WHO), the cumulative global HIV cases in 2020 reached 37.7 million people, with 1.5 million new HIV cases, and 680,000 deaths attributed to the disease. The high prevalence of HIV in the Southeast Asia region places Indonesia on alert for the transmission and spread of the virus [10]. Regardless, data on HIV/AIDS cases are expected to continue increasing annually, with the highest number of cases from 2017 to 2019 concentrated on Java Island [11].

Generally, HIV disrupts the human system, resulting in weakened immune defenses upon infection compared to uninfected people. HIV places PLWHA at risk for opportunistic infections, various microorganisms (fungi, other viruses, bacteria, and other parasites) seize the opportunity to infect when the immune system is compromised [12]. These chronic diseases can be managed by antiretroviral (ARV) therapy, which must be taken throughout life Antiretrovirals. Non-adherence with ARV therapy can make PLWHA resistant to ARV, often known as ARVs; these are the only drugs that offer significant benefits by suppressing the growth of HIV within the body. By decreasing the replication of this virus, ARVs ensure the maintenance of the human immune system (CD4) [13].

In nursing care, the psychospiritual aspect is often overlooked. However, experts [14] state that a nurse’s goal is to enhance the adaptation of individuals and groups across four adaptive modes (physiologic, self-concept, role function, and interdependence modes), thereby contributing to an individual’s quality of life, complex health, and honorable end-of-life. Living with HIV involves managing various disease-related stresses and life events. Consequently, distress tolerance may provide an individual-based functional context to qualify the well-established relationship between significant life events and essential psychosocial variables in managing ARV therapy for PLWHA patients [15,16]. According to research, distress tolerance is the ability to tolerate various kinds of emotional experiences and physical conflict, such as unfavorable affective conditions, physical discomfort, etc. [17].

Since the mid-20th century, research has indicated a relationship between religiosity and spirituality and health [18,19]. This signifies that healthcare providers should possess the necessary skills to detect patients’ spiritual needs, stress, and well-being to take note from nursing and apply a holistic, comprehensive approach that provides physical as well as spiritual care [20]. The development of an individual’s spiritual well-being correlates with other spiritual dimensions, including their understanding of religion and its progression in their daily lives. Spiritual well-being as an independent concept within the context of counseling and guidance services continues to evolve within human beings [21]. Spirituality is an inherent characteristic of people and a potent source of healing. The spiritual dimension provides meaning to the relationship with God, oneself, and the surrounding world [22].

Spiritual well-being significantly influences reducing the risk of health issues in the present and beyond [23]. Distress tolerance has an effect on improving the health behavior of PLWHA [24]. Symptoms of HIV withdrawal from medication or non-adherence with taking ARV that can be felt by PLWHA are the increase in HIV and the emergence of various disease developments in the body [1]. Based on researchers’ searches in the literature, there are many studies on variables related to ARV therapy adherence, including a previous study in Papua in 2021 which shows several variables related to ARV therapy adherence, but no study was found on spiritual well-being and distress tolerance on ARV therapy adherence. The purpose of this study is to describe the determinants of adherence and analyze the relationship between distress tolerance and spiritual well-being on ARV therapy adherence in PLWHA. Healthcare services to PLWHA should consider these factors to increase adherence among patients and inadvertently heighten mortality risk.

## 2. Materials and Methods

### 2.1. Research Design

This research used a quantitative approach with a cross-sectional design.

### 2.2. Data and Research Samples

The study was conducted from January to March 2023 in the population of People Living with HIV/AIDS (PLWHA), especially in South Lampung Regional General Hospital, which is the only province with an entry point for the spread of HIV on the island of Sumatra from the island of Java via land routes in Indonesia, using the consecutive sampling technique. Based on the highest number of routine patients throughout the year, in December 2022 there were 191 PLWHA in outpatient care. This study recorded a sample of 129 PLWHA using inclusion criteria which included PLWHA who visited the Voluntary Counseling and Testing (VCT) outpatient clinic at the South Lampung Regional General Hospital, Indonesia; those whose information is recorded in the medical record; those who adhere to a religion recognized by the Indonesian state; individuals who are able to read and write; and those who are willing to participate as respondents by signing informed consent. Primary data were collected directly by researchers using a questionnaire containing questions and statements.

### 2.3. Research Variables

The research variables consist of the dependent variable, namely adherence to ARV therapy in PLWHA, while the independent variables are distress tolerance and spiritual well-being. Data collection involved three questionnaires with a construct that is declared reliable if Cronbach’s alpha > 0.60 [25]. Firstly, distress tolerance was assessed using the Miller–Smith Rating Scale for Stress Tolerance (MSRS-ST), which yielded a Cronbach’s alpha value of 0.671 following validity testing by Khairiyah in 2008. Secondly, spiritual well-being was evaluated via the Spiritual Well-Being Scale (SWBS), which attained a Cronbach’s alpha value of 0.911. This scale was adapted from the questionnaire developed by Paloutzian and Ellison in 1982, previously used by Utama in 2018 and had been translated into Indonesian. Finally, adherence to ARV therapy was assessed using the Medication Adherence Rating Scale (MARS), which showed a validity of 0.779 with Cronbach’s alpha test, adopted from a questionnaire by Thomson et al. (2000) [26]. The instrument had been licensed and formerly utilized in research by Sianturi et al. in 2019 [27]. For the independent variable, there are two variables, namely tolerance to stress, which are grouped into two categories: those with an MSRS-ST score of <43 are categorized as respondents with good distress tolerance and if the MSRS-ST score is ≥43, they are categorized as respondents with poor distress tolerance. The spiritual well-being variable is categorized as good spiritual well-being if the total SWBS score is ≥79 and poor spiritual well-being if the total SWBS score is <79. The dependent variable, namely Adherence with ARV therapy, is cut to adherence with ARV therapy if the MARS score is ≥6 and non-adherence with ARV therapy if the MARS score is <6. Confounding variables included age, sex, marital status, education, and employment, and were defined as follows: age was categorically defined as <34 versus ≥34 based on the median of the sample with 19–59 years old according to the number of years of life the respondent; sex was categorically defined as male versus female based on gender recorded in the state’s legal documents; apart from that, marital status was broken down into three categories, namely married, never married, and divorce, according to the marital status records in the state’s legal documents; education was categorically defined as basic education if the respondent had an elementary school or junior high school education versus further education, which was if the respondent had a senior high school or college education based on Indonesian education formal level. The final confounding variable is employment, which was reduced to working if the respondent was still actively working and not working if the respondent was not actively working to earn income.

### 2.4. Data Analysis

The data analysis for this research included univariate, bivariate, and multivariate analyses. The univariate analysis took the form of frequency distribution tables, capturing various categorical data. The bivariate analysis also consisted of categorical data and utilized simple logistic regression to determine the relationship between the independent and dependent variables using a 95% confidence interval (CI) and a significance level of *p* < 0.05. Multivariate analysis was employed to identify the most influential variables in this research. Since the independent, confounding, and dependent variables were all categorical, multiple logistic regression analysis was used.

### 2.5. Ethical Considerations

This research obtained approval and was ethically reviewed by the Faculty of Nursing, University of Indonesia, with the number: Ket-11/UN2.F12.D1.2.1/PPM.00.02/2022.

## 3. Results

### Respondent Characteristics

The respondent characteristics consisted of age, gender, education level, occupation, religion, marital status, distress tolerance, spiritual well-being, and ARV therapy adherence, as shown in Table 1, Table 2 and Table 3 below.

The statistical test results demonstrate a relationship between distress tolerance and adherence to ARV therapy in PLWHA (*p*-value = 0.02 with α = 0.05). The statistical test results indicate a relationship between spiritual well-being with adherence to ARV therapy in PLWHA (*p*-value = 0.036 with α = 0.05).

The final results of multiple logistic regression modeling by entering univariate variables as covariates (age, sex, marital status, education, and employment). Based on the OR values contained in the table above, the distress tolerance variable had the greatest influence on adherence to ARV therapy among PLWHA, at a *p*-value of 0.01 and an OR of 5.004. The largest OR value obtained is 5.004, which means that distress tolerance has a 5.004 times chance of affecting adherence to ARV therapy in PLWHA.

## 4. Discussion

### 4.1. Distress Tolerance against Compliance with ARV Therapy

This research found a relationship between distress tolerance and adherence to ARV therapy in PLWHA. The distress tolerance variable had the greatest influence on adherence to ARV therapy among PLWHA. Research has demonstrated that even missing ARV medication for just a week can have fatal consequences for the health of PLWHA [1]. People with low distress tolerance and greater life experience are more likely to have poorer adherence to antiretroviral treatment. Generally, stressful life events, depression, hopelessness, anxiety, low social support, and low HIV knowledge were associated with poor adherence. The distress tolerance refers to the actual capacity or ability to tolerate emotional experiences and physical discomforts, such as negative affective conditions, physical distress, etc. [15].

In a previous study, distress tolerance significantly moderated the relationship between major life events in the past 6 months and relevant outcomes in the HIV population, namely depressive symptoms, substance use, and alcohol and cocaine use [15]. Low distress tolerance combined with a high frequency of major life events significantly increases the strength of this relationship. These results are consistent with and extend previous findings that specific stress levels interact with stress tolerance to predict subsequent panic onset [28,29]. Furthermore, the presence of a moderate association with depression has particular clinical relevance in HIV due to evidence of its propensity to predict accelerated disease progression [30] and death [31]. Depression in people living with HIV also interferes with the adaptive management of the disease through its association with higher rates of drug use [24,32] and poorer adherence to antiretroviral treatment [33].

Another study reported that trauma, stressful events, and post-traumatic stress disorder accounted for about 27% of the variance in health-related functioning in a large sample of HIV-seropositive patients. This suggests that life stress and the associated emotional distress are strongly related to physical functioning [34]. Irrespective of the mechanism underlying this association, this research suggests that the evaluation of stress tolerance is well-indicated to support adaptive HIV management, as well as to inform treatment. Patients with low stress tolerance and high survival events have more reasons to skip their antiretroviral drugs. Poor adherence has been previously associated with stressful life events, depression, hopelessness, anxiety, lower social support, and lower levels of HIV knowledge [35,36]. However, this may be the first research to establish an association between adherence and stress tolerance in HIV management. Adherence to antiretroviral treatment is critical to the effective management of HIV [37,38].

A state of stress is an internal condition caused by the physical needs of the body (unhealthy conditions and physical exercise) or by environmental and social conditions that are considered potentially detrimental, uncontrollable, or excessive, and the ability to overcome them [39]. Distress tolerance towards pressure may provide an individual-based functional context to qualify the established relationship between major life events and important psychosocial variables in HIV management [16]. Similarly, stressful life events have been shown to cause low adherence to retroviral drug use [35,40,41].

### 4.2. Relationship between Spiritual Well-Being and Compliance with ARV Therapy

This analysis showed a relationship between spiritual well-being and adherence to ARV therapy in PLWHA. In other research, spiritual well-being has been closely related to mental health and can increase patients’ enthusiasm and optimism about their illness [21]. Spiritual well-being is linked to growth in the spiritual dimension, along with an appreciation for and the development of people’s religion [42]. Research has also detected a higher religious score following the best treatment for adolescents with HIV, identifying that young people with HIV/AIDS are more likely to ask themselves, “Has God abandoned me?” In this case, adolescent spirituality is associated with reduced anxiety and depression, along with better adaptation to chronic illness [43]. Spiritual well-being is an independent concept in the context of counselling and guidance services that develop within each individual [21]. This is due to the complexity and multi-dimensional nature of spirituality, as each person experiences, interprets, and comprehends a unique perception of the phenomenon [44].

Spiritual well-being is an independent concept that develops in each individual through personal service and guidance according to their respective concepts [45]. PLWHA still requires an approach to achieve good spiritual well-being, thereby demonstrating a strong desire for healing. People in this group make continuous efforts to undergo therapy, attend clinics regularly to obtain medication, adhere consistently to ARV treatment without interruption, maintain healthy lifestyles, and remain faithful to their partners. These behaviors lead to an improved life and enhance the goal of recovery [46]. Spiritual well-being is also an independent concept in the context of counseling and guidance services, which continues to develop throughout the lives of individuals [21].

Meanwhile, negative coping styles associated with less devoutness refer to a lack of attachment to God, difficulties in finding meaning in life, and spiritual struggles, defining stressors as punishment from God or acts of evil [47]. Conversely, conventional belief in God as a source of comfort, support, and assistance to cope with stress appears to be associated with better adherence, while a fundamentalist belief in God’s power to heal prevents some patients from continuing pharmacological treatment [48]. In Ethiopia, the healing potential of holy water is associated with non-adherence to ARV [49]. Another study conducted in Tanzania found that most participants taking antiretrovirals sought herbal remedies for HIV by visiting spiritual leaders [50]. The presence of religion also displaces belief in Mapuche healers, with healers being the main actors in the health recovery process [7].

In order to address and utilize patients’ spirituality as a supportive mechanism for therapeutic adherence, a specialized department should be established to manage the integration of spirituality with conventional treatment in the hospital [51]. Spiritual therapy could be provided by volunteers and offered free of charge to patients and staff. This has been demonstrated by research, where patients showed a willingness to discuss religious matters with their psychiatrists [52]. They also become more engaged and have better relationships with doctors who possess a greater understanding of specific religions [53], thereby promoting cultural sensitivity and offering assistance through the healthcare system. Notably, the support provided through religious affiliation or specific practices, such as prayer, has a positive impact on adherence to ARV in HIV/AIDS management [54].

### 4.3. The Most Influential Variable

This research demonstrated that the distress tolerance variable had the most significant influence, regardless of the context, among individuals living with HIV. Stress is a condition that induces physical changes (biochemical, neurohormonal, and immune), as well as psychological alterations (cognitive and behavioral) [55]. Excessive or chronic stress can have adverse effects, leading to the emergence of mental health problems, such as psychological disorders, emotional instability, and even suicide, along with physical disorders, including chronic conditions, like cardiovascular, musculoskeletal, gastrointestinal issues, cancer, and weakened immunity [56]. Generally, people with HIV struggle with managing a chronic illness, as well as reporting a high incidence of other life stressors [34]. Distress tolerance is the level of actual ability or feeling to endure emotional experiences and physical conflicts, such as negative affective conditions, physical distress, etc. [17]. The observed relationship between stress tolerance and reported HIV symptoms indicates that those with low stress tolerance will be more sensitive and prone to the onset of HIV-related symptoms [15].

According to psychopathology literature, the impact of stress on clinical outcomes may depend on the individual’s level of stress tolerance. Distress tolerance is a high-level emotional variable that influences the capacity of an individual to experience and withstand emotional discomfort [57]. Meanwhile, individual spiritual well-being also significantly influences personal life by reducing the risk of health problems in the present and beyond [23]. According to research, the role of nurses is to enhance health, improve quality of life, and facilitate peaceful death [14]. Distress tolerance relates to improved health behavior of PLWHA in their daily lives, particularly in ARV therapy adherence, thereby enhancing their quality of life. Individuals with high stress tolerance can withstand and adapt well to various stressors, whereas those with low tolerance are unable to adapt to stress effectively, thereby becoming susceptible to experiencing distress, resulting in negative consequences on physical, mental, spiritual health, and overall quality of life [58].

### 4.4. Limitations of the Study

Some participant candidates declined participation, due to factors unrelated to the research itself, but those who participated in this study still represent the population in this study. The research results obtained may not reflect the beliefs and attitudes reported in younger samples or PLWHA in other areas considering that the data in the study were collected from a single site. In this study, to minimize the potential for response bias and social desirability, an anonymous survey was conducted on all respondents.

### 4.5. Suggestion

Many health facilities do not pay attention to distress tolerance and spiritual well-being in PLWHA patients. Based on this research, by paying attention to these two aspects, we can reduce the risk of non-adherence with ARV therapy among PLWHA in hospitals and other health facilities. Apart from that, education can also be developed about the factors that influence adherence to ARV therapy in PLWHA in an effort to increase adherence to ARV therapy in PLWHA.

## 5. Conclusions

In conclusion, a significant effect was observed in the relationship between distress tolerance and spiritual well-being regarding ARV therapy adherence in PLWHA. Among the variables studied, the distress tolerance variable exerted the most dominant influence on therapy adherence. As a result, both distress tolerance and spiritual well-being in PLWHA should be highly considered in standard clinical care to effectively reduce the risks of non-adherence to ARV therapy and mortality.

## Figures and Tables

**Table 1 healthcare-12-00839-t001:** Frequency Distribution of Respondents based on Age, Sex, Marital Status, Education, Distress Tolerance, Spiritual Well-Being, and Adherence with ARV Therapy in PLWHA (*n* = 129).

Variable	*n*	%
Age (Median: 34; Mean: 34.6; Min–Max: 19–59)		
Age > 34 years	67	51.9
Age ≤ 34 years	62	48.1
Sex		
Male	78	60.5
Female	51	39.5
Marital Status		
Married	53	41.0
Never married	50	38.8
Divorce	26	20.2
Education		
Basic Education	9	7
Further Education	120	93
Employment		
Working	96	25.6
Not Working	33	74.4
Distress Tolerance		
Good Distress Tolerance	56	43.7
Poor Distress Tolerance	73	56.3
Spiritual Well-Being		
Good Spritual Well-being	67	51.9
Poor Spritual Well-being	62	48.1
Adherence		
Adhered to ARV Therapy	86	66.7
Non-Adhered to ARV Therapy	43	33.3

**Table 2 healthcare-12-00839-t002:** The Relationship between Distress Tolerance and Spiritual Well-Being on ARV Therapy Adherence in PLWHA (*n* = 129).

Variable	*p*-Value
Distress Tolerance	0.01
Spiritual Well-Being	0.03
Age	0.69
Sex	0.52
Marital Status	0.41
Education	0.47
Employment	0.84

*p*-value: <0.05.

**Table 3 healthcare-12-00839-t003:** Final Model Selection of the Most Dominant Variable Related to Adherence with ARV Therapy in PLWHA (*n* = 129).

Variable	B	OR	*p*-Value
Distress Tolerance	1.610	5.004	0.01
Spiritual Well-Being	0.344	1.411	0.406

B: Multivariate regression model parameter matrix, OR: Odd Ratio, CI: Confidence Interval, *p*-value: <0.05.

## Data Availability

The data presented in this study are available on request from the corresponding author.

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
