# Peer review of "The Relationship between Distress Tolerance and Spiritual Well-Being towards ARV Therapy Adherence in People Living with HIV/AIDS"

_healthcare, 2024, doi:10.3390/healthcare12080839_

Round 1

Reviewer 1 Report

Comments and Suggestions for Authors

The authors are commended for their examination of stress tolerance and spiritual well-being as it relates to ARV adherence in adults with HIV. Ability to cope with daily life stressors in addition to HIV are important, which may include reliance of spiritual and/or faith beliefs as a source of strength.  While there are no major concerns identified, and this manuscript has the ability to contribute to the extant literature, there are many, many minor concerns or omissions that need to be addressed as follows:

Concern:

1)   End of intro: Please state in a summary sentence what gap in the literature your research intended to address.

Minor issues:

1)    Title: “With” does not need to be capitalized.

2)    Abstract: Anti-Retroviral does not need to be capitalized. Further, it is accepted in the literature that the hyphen also is not needed. Please replace with antiretroviral.

        2a) Cross-Sectional does not need to be capitalized.

3)    Abstract, line 16: Please specify if questionnaires were self-completed or interviewer administered. It appears from the methods these may have been interviewer-administered.

4)    Abstract, last sentence, recommend rephrasing “the potential for mortality” to read: “inadvertently heighten mortality risk.”

5)    Introduction, Line 31: decrease in “national development” – what does this mean exactly?

        5a) please also end this sentence to specify which nation you are speaking about.

6)    Intro, Lines 41-46: “Infections among PLWHA are often referred to as opportunistic infections, various microorganisms (fungi, other viruses, bacteria, and other parasites) seize the opportunity to infect when the immune system is compromised [12].” Recommend rephrasing the introductory phrase ““Infections among PLWHA are often referred to as opportunistic infections” to read “HIV places PLWHA at risk for opportunistic infections…”

7)    Intro, line 58: what are the 4 adaptive modes? Please list.

8)    Intro, line 73: “nursing?” you mention providers earlier in the sentence. Do you simply mean “holistic, comprehensive approach”? or do you mean to take a note from nursing and apply a holistic, comprehensive approach?

9)    Data and Research Samples: line 90: VCT is not defined.

10) MSRS-ST, line 100: reported Cronbach alpha 0.0671: is this a typo? Is it really that low or should it be .671? Validity is below .70, which is typically considered marginally acceptable (i.e., moderate validity). You will need to include this as a potential limitation to first acknowledge that, defend its use, and that you cannot say how it may have performed differently in your sample (better or worse).

11)  MMAS, line 106: Same issue. Validity is below .70, which is typically considered marginally acceptable (i.e., moderate validity). You will need to include this as a potential limitation to first acknowledge that, defend its use, and that you cannot say how it may have performed differently in your sample (better or worse).

12)  Lines 109-110: “<43 and tolerance to bad stress if the MSRSTS score is >43,”…is this supposed to read “low/poor tolerance to stress”? Good tolerance versus bad tolerance instead of bad stress? Please clarify this sentence. Also remove the comma at the end.

13) Lines 113-114: please cite a reference for the use of the MMAS criterion cut score.

14)  Lines 114-115: When introducing your confounding variables, perhaps, first list, then define each, which would allow you to make this paragraph more concise and eliminate the repeated “apart from that.” For example, rewrite the first sentence to read: “Confounding variables included age, sex, martial status, education and employment defined as follows: Age was categorically defined as < 34 versus > 34 based on (mean, median, mode of the sample?); biological sex was categorically defined as…: marital status was categorically defined as…,” etc.

15) Line 115: please defined how the age categorization < 34 vs >34 was determined. Was it the mean age of your sample? The mode? And is it < 34 or >34? Which category includes age 34?

16) Line 118: is “divorced” included in “unmarried”, i.e., never married and divorced are included together?

17) Analysis, line 128: “generation of insights”? What does this mean? I don’t think this sentence is necessary. It is understood that SPSS is a computer-generated analytic software.

18) Results Table 1: The first and second columns need to be switched, with the second column (variables) needs to be listed first, following by the categorization of each listed variable.

18a) please also provide the mean age and age range of your sample to aid in interpretation of your data.

        18b) “Basic” and “Further” education do not match your categorizations for the education variable. Please explain where you list your definitions of education, how those were reduced to this dichotomous variable, or provide the data in the table for all categories (preferred).

19) Results Lines 146-151: This summary of the table 1 data is not needed as it is redundant with what is presented in the table without adding any additional interpretation.

20) Results Line 169: p-value <0.000 – this actually needs a 1 added either after the 3rd zero, or replacing the 3rd zero, regardless of what the SPSS printout shows, it is understood a p-value is truly never 0.000…

21) Results, lines 171-174: the last sentence should be moved to follow the OR presented in line 169.

22)  Results, Line 180: “greater life experience” – do you mean older age or do you mean more negative life events? This can be interpreted more than one way. Please clarify intended meaning.

23)  Discussion, line 181: “motivations” – while the intention is understood, the reader has to remember your results to recall that this reflects poorer functioning and really means greater “need.” Recommend using “need” instead of “motivation”, especially since motivation also implies desire. As worded: “People with low distress tolerance and greater life experience are more likely to have

motivations for adhering to antiretroviral treatment.” Recommend rewording to instead read: “…are more likely to have poorer adherence to antiretroviral treatment.”

24) Discussion, first para: in general, this first paragraph would benefit from revision to improve reading

flow. It takes more than one read to grasp intended meaning.

25) Discussion, Line 228, “person” – do you mean “present”? Use of person here is not clear.

26) Discussion, Line 240: “obedience” is pejorative. Are you referring to ARV adherence here or to religiosity? If ARV adherence, please use the word adherence here. If you are referring to religiosity, perhaps use the wording “less devout”.

27) Discussion, Line 247-252: A caution here should be added. In the US, it is well understood for populations who engage in cultural practices using herbs that many of these herbal remedies actually interact with ARV regimens, decreasing their efficacy, risking the development of viral resistance and potential risk for liver toxicity, and as a result are strongly cautioned against, having to balance cultural practices with Western medicine. Please do a brief literature search to review these concerns and add a statement here that includes the needed caution so that an uninformed reader, including potential providers, do not assume herbal remedies are safe for those prescribed ARVs because of accepted cultural practice for those who do not have HIV.

28)  Discussion, Line 269: “mental problems” – please rewrite to read “mental health problems.”

29) Discussion, Line 286: Please open this sentence with “Better…” and revise to read: “Better distress tolerance relates to improved health behavior…”

30)  Discussion, Line 291: add to the end of this sentence: …experiencing distress, resulting in negative consequences on physical, mental, and spritual health, and overall quality of life.”

31) Limitations: Define ODHA

32) Line 302: “non-compliance” – please use “non-adherence.”

33) Line 311: “should have been” – please reword to read “should be highly considered in standard clinical care to effectively…”

34) Minor typographical, grammatical, word use and word agreement issues throughout that require careful review and editing by a more experienced published author fluent in written English before resubmitting.

Comments on the Quality of English Language

I believe this manuscript has the ability to contribute to the extant literature but needs editorial assistance. The content is there but there also are a lot of minor word-use, grammatical and punctuation edits needed in addition to a final review by an experienced English-proficient reader/author/editor. As written, the authors would benefit from mentorship of a more experienced published author to finesse the manuscript before resubmitting.

Author Response

We thank the editor and reviewers for their thorough reading of our manuscript and their comments and suggestions that helped us to improve our manuscript entitled The relationship between distress tolerance and spiritual well-being towards ARV therapy adherence in People Living With HIV/AIDS. As indicated below, we have tried to do our best to respond to all the points raised. Please contact me if you need any further information.

Reviewer 2 Report

Comments and Suggestions for Authors

Dear authors,

You explored theme is really important, but in my opinion you could have explored more your data. For example, you can analysis possible associations of data presented in Table 1 with ARV adherence and with the scales.  Perhaps you will get interesting results. For example, adherence can be associated with gender and spiritual well being not present any association with gender. Therefore, please, explored more your data!

Another  point is about the sample size calculus. What criteria did you apply here (CI,...)? It should be more clear how you get this goal of 129 people and the reason why you chose this hospital.

Bellow I list some few suggestions:

1 – Introduction is very well written, but without present the main goal of the study. However, I could not see the originality of this study. Everybody knows that distress and well being are essential to keep ARV adherence. How many previous studies have been done in this subject? Have you done a literature mapping before? What is the main goal of this study? These information need to be clear in the text. 

2 – ARV adherence and the treatment abandonment should be conceptualized.

3 – First paragraph of the discussion should contain the main results obtained in the study.

4 – “In this research, distress tolerance significantly moderated the relationship between 186 major life events in the past 6 months and relevant outcomes in the HIV population, 187 namely depressive symptoms, substance use, and alcohol and cocaine use [24].” This sentence should be rebuilt. I suggest you to change “this research” for “Previou study”.

Line 170, what do you mean by "as 1.411 respondents"?

Comments on the Quality of English Language

Punctuation and grammar review should be applied in the manuscript.

Author Response

(The authors gave the same response as above.)

Reviewer 3 Report

Comments and Suggestions for Authors

The paper is certainly interesting and considers an important and often overlooked aspect of the life of PLWHA. The manuscript is written in good English and, although some redundant parts, the paper is overall very readable.

The introduction was exhaustive in clarifying the concept of spiritual well- being. That said, I was wondering if there were non-believers among the PLWHA or only people with a religion were considered (if so, please specify in the inclusion criteria) and if there were people with different religions among the people included in the study.

Lines 61-63 and 66-69 are basically expressing the same concept with almost the same words, please eliminate the redundant parts.

Comments on the Quality of English Language

English Language requires minor editing, but is overall good and readable.

Author Response

(The authors gave the same response as above.)

Round 2

Reviewer 1 Report

Comments and Suggestions for Authors

The authors are commended for their efforts to respond to this reviewer’s previously provided feedback, however several issues remain that should be addressed at an editorial level. This manuscript has the ability to contribute to the extant literature, but the one major and few minor remaining concerns need to be addressed are as follows:

Major Concern:

1)   This manuscript still needs significant editorial assistance. As previously noted, as written, the language use, grammatical issues and/or writing style concerns remain, including redundancies, vague word choices, and some statements that seem tangential at best; these concerns also are present in the new text following revision. As previously noted, the content is there, but the writing style simply is not fluid, lacks clarity, and reflects lack of writing experience expected for publication that unfortunately negatively affects the reading flow and the reader’s ability to understand the authors’ intended meaning. It is again recommended the authors solicit a final review and editorial feedback from by an experienced English-proficient reader/writer/published author before resubmitting a revised final manuscript. Although the authors state this was done, it does not appear so.

Minor issues:

1)   Some typos and punctuation issues remain.

2)   p-value 0.00 is presented in Table III and in the text below it. It should be >0.01 or >0.001, whichever you choose.

3)   Limitations: I get that you are trying to say that your study is limited by some participant candidates declined participation due to factors unrelated to the research itself, and that you believe those who did participate are still representative, but this paragraph is a great example of lacking clarity, vague word choices and redundancy. Further, there are additional limitations to report: Limitations inherent in self-report measures, including potential response-bias and social-desirability, etc. Given your data were collected from a single site, your results may not reflect the beliefs and attitudes in a younger sample, or PLWHA in other regions, etc. These should be added. You should also consider whether your univariate findings were lost or strengthened in multivariable analyses that included covariates…

Again, the content is there. The writing needs improvement.

Comments on the Quality of English Language

The authors need extensive editorial assistance to improve the readability/flow of the manuscript. As noted, the content is there, it just needs work to improve the fluidity/presentation of intended thoughts. It requires more editorial assistance than appropriate for a reviewer to provide. I really believe this manuscript has the ability to contribute once appropriately revised.

Author Response

dear reviewer 1, 

i attach it below. 

thankyou

Reviewer 2 Report

Comments and Suggestions for Authors

Dears authors, 

Is there any previous study addressing the same subject of yours? Have you searched it in the literature data basis?  If you checked, please, I suggest you to indicate in the introduction the possible previous study and the difference between your study and the previous ones. If you haven't found any previous study, state it too. It is necessary to reinforce the originality of the study. Science are here to close gaps!

- Still the sample size calculus not clear. How did you reach the goal of 129 patients? How many patients are assigned in VCT? You have to prove employing appropriated sampling calculus that these 129 are representative of all patients assigned in VCT. If you don't prove it, your results will not represent this population.  In the text sounds that you were to VCT and chose 129 patients and that is it.

(Please, sample size is the main component of a study. Detail each step clearly!)

In discussion you should mention where studies were done (e.g.: Previous study in Albania showed....)

Author Response

dear reviewer 2

i attach it below 

thankyou
